# Genetical Signature—An Example of a Personalized Skin Aging Investigation with Possible Implementation in Clinical Practice

**DOI:** 10.3390/jpm13091305

**Published:** 2023-08-25

**Authors:** Ramune Sepetiene, Vaiva Patamsyte, Paulius Valiukevicius, Emilija Gecyte, Vilius Skipskis, Dovydas Gecys, Zita Stanioniene, Svajunas Barakauskas

**Affiliations:** 1Laboratory of Molecular Cardiology, Institute of Cardiology, Lithuanian University of Health Sciences, Sukileliu St. 15, LT-50162 Kaunas, Lithuania; vaiva.patamsyte@lsmuni.lt (V.P.); emilija.gecyte@lsmuni.lt (E.G.); vilius.skipskis@lsmuni.lt (V.S.); dovydas.gecys@lsmuni.lt (D.G.); zita.stanioniene@lsmuni.lt (Z.S.); 2Abbott GmbH, Max-Planck-Ring 2, 65205 Wiesbaden, Germany; 3Faculty of Medicine, Medical Academy, Lithuanian University of Health Sciences, Mickeviciaus 9, LT-44307 Kaunas, Lithuania; paulius.valiukevicius@lsmuni.lt; 4LTD Medicina Practica Laboratorija, Laisves Pr. 78B, LT-05263 Vilnius, Lithuania; svajunas@medicinapractica.lt

**Keywords:** skin aging, polymorphisms, laboratory analysis, personalized medicine

## Abstract

We conducted a research study to create the groundwork for personalized solutions within a skin aging segment. This test utilizes genetic and general laboratory data to predict individual susceptibility to weak skin characteristics, leveraging the research on genetic polymorphisms related to skin functional properties. A cross-sectional study was conducted in a collaboration between the Private Clinic Medicina Practica Laboratory (Vilnius, Lithuania) and the Public Institution Lithuanian University of Health Sciences (Kaunas, Lithuania). A total of 370 participants agreed to participate in the project. The median age of the respondents was 40, with a range of 19 to 74 years. After the literature search, we selected 15 polymorphisms of the genes related to skin aging, which were subsequently categorized in terms of different skin functions: *SOD2* (rs4880), *GPX1* (rs1050450), *NQO1* (rs1800566), *CAT* (rs1001179), *TYR* (rs1126809), *SLC45A2* (rs26722), SLC*45A2* (rs16891982), *MMP1* (rs1799750), *ELN* (rs7787362), *COL1A1* (rs1800012), AHR (rs2066853), *IL6 (rs1800795), IL1Beta* (rs1143634), *TNF-α* (rs1800629), and *AQP3* (rs17553719). RT genotyping, blood count, and immunochemistry results were analyzed using statistical methods. The obtained results show significant associations between genotyping models and routine blood screens. These findings demonstrate the personalized medicine approach for the aging segment and further add to the growing literature. Further investigation is warranted to fully understand the complex interplay between genetic factors, environmental influences, and skin aging.

## 1. Introduction

“Aging and beauty” are two frequently juxtaposed concepts, where “healthy and beautiful” serve as synonyms. Over time, doctors and scientists have focused on creating the elixir of youth or a formula for immortality that would work for all of humankind. The past century has witnessed significant advancements in genetic discoveries and the elucidation of molecular secrets, which have recently culminated in a transformative impact on the field of medicine, particularly in the realms of treatment advancements and personalized approaches [1,2]. Moreover, a comprehensive understanding of intrinsic and extrinsic factors impacting the body has played a crucial role in the interesting findings of genetic pleiotropy and clear mechanisms of skin aging, which is an inevitable process, and all attempts to search for individualized diagnostics and treatment procedures could give the best results [3]. When examining variations in biological and social age across different racial or ethnic groups, investigators found that the influence of intrinsic factors supported the concept of a personalized medicine approach to aging, without exceptions for skin aging [4].

Considering the current trends toward longevity and the prioritization of a precision medicine approach, research studies are ongoing on a very wide spectrum of medicine topics, such as genomic correlation with cardiovascular and neurogenerative disease [5], as well as various prediction models in cancer and rare disease treatment [6]. Highly successful examples of personalized medicine implementation include the discovery of genetically mediated pharmacokinetics of drug-metabolizing enzymes [7] and the treatment of chronic myelogenous leukemia with Imatinib [8]. The progress and availability of genotyping and sequencing techniques for individuals, particularly imaging technologies and blood-based clinical assays, combined with their accessibility and inexpensiveness, allows for the continuous monitoring of an individual’s health status and health-related concerns [9].

Last decade, significant advancements in genome-wide association studies (GWASs) focusing on skin aging made considerable progress in analyzing various mechanisms and identifying the underlying reasons for skin aging [3,10,11]. Building upon these works, we conducted a collaborative cross-sectional research study to create the groundwork for personalized solutions and develop a reference skin. This test utilizes genetic and general laboratory data to predict individual susceptibility to weak skin characteristics, leveraging the research on genetic polymorphisms related to skin functional properties.

## 2. Materials and Methods

### 2.1. Study Design

Between 2019 and 2021, a cross-sectional study was conducted in a collaboration between the Private Clinic Medicina Practica Laboratory (Vilnius, Lithuania) and the Public Institution Lithuanian University of Health Sciences (Kaunas, Lithuania). Informed consent was obtained from each participant recruited into the study. The study protocol was approved by the Regional Medical Research Bioethics Committee: BE-2-53 (10 June 2019).

### 2.2. Study Population and Exclusion Criteria

The study population comprised healthy individuals aged ≥18 years who signed a written informed consent form, which resulted in 370 participants. SNPs were selected for laboratory analysis with respect to possible PCR limitations, and information was collected from studies that have been published within the last 10 years.

Blood samples were collected using the standard venepuncture procedure, which involved obtaining two tubes (1 EDTA tube for the complete blood count and DNA extraction and 1 tube with gel to accelerate the separation of serum for immunochemical assays). Subjects who had a concomitant medical illness or anyone with ongoing ailments, such as viral or bacterial infections, that could potentially affect the results were excluded. A skin condition self-assessment form was filled out by all participants. Fifteen questions were grouped into 5 different groups in terms of skin reaction or resistance to the most common environmental and individual factors: hydration, inflammation, elasticity [11], mechanical resistance, and other (skin pigmentation, food supplemental use, and regular beauty treatments). The assessment of the skin condition of the study participants was subjective, they answered the standard questions based on their complaints, changes in the appearance of the skin, and their subjective opinions related to their perception and notes. We did not find any significant associations in the self-assessment data between all laboratory results. Thus, this part was excluded from further analysis. To assess the objective appearance and condition of the skin, a different study design would have had to be chosen, i.e., a dermatologist would have had to provide standardized conclusions and evaluation results via objective means (dermatoscope, histological examination, etc.). This was the first stage of the study to obtain initial results to explore further research with an extended team of investigators.

### 2.3. Single-Nucleotide Polymorphism Detection

Genomic DNA was extracted from whole blood samples using a commercial PureLink Genomic DNA kit (Invitrogen, Thermo Fisher Scientific, Bleiswijk, The Netherlands) according to the manufacturer’s recommendation and quantified using a NanoDrop 2000 (Thermo Fisher Scientific, Bleiswijk, The Netherlands) spectrophotometer. SNP genotyping was performed on a 7900 HT Real-Time PCR system using TaqMan (Life Technologies, Thermofisher Scientific, Bleiswijk, The Netherlands) chemistry under standard conditions. Primary SNP selection was obtained using a scientific literature search with the keywords: “GWAS, polymorphisms, skin aging, personal skin care” on the PubMed NCBI database website (https://www.ncbi.nlm.nih.gov/pmc, accessed on 2 June 2017) with filters for “full text, author manuscript, open access, not older than 5 years”. A total of 67 articles were found. Sequencing-only studies were excluded from the analysis, and instead, SNPs were selected for analysis using real-time genotyping. To ensure the selection of relevant SNPs, a search was conducted in the SNP database (https://www.snpedia.com/, accessed on 2 June 2017). The aim was to identify SNPs that were most commonly found in European populations, with a minimum occurrence of 1.00% for the minor genotype. Based on these criteria, 16 SNPs were chosen for the final laboratory analysis. However, one SNP (rs35652124) was excluded from further investigation due to primer synthesis defects that rendered it unsuitable for analysis. All 15 SNPs investigated and analyzed are presented in Table 1. This part of the study was performed in the Laboratory of Molecular Cardiology at the Lithuanian University of Health Sciences.

### 2.4. Complete Blood Count and Biochemical Analysis

Clinical laboratory testing was performed in a certified clinical diagnostics laboratory “Medicina Practica Laboratorija” (Vilnius, Lithuania). The complete blood count was determined on a Sysmex XT-1800 analyzer using a Roche Diagnostic kit according to standard procedures. Biochemical and serological blood testing was performed on a Cobas 6000 analyzer and included the following: C-reactive protein (CRP), aspartate aminotransferase (AST), alanine transaminase (ALT), gamma-glutamyl transferase (GGT), alkaline phosphatase, pancreatic amylase, a lipid panel (total cholesterol, triglycerides, low-density lipoprotein (LDL), high-density lipoprotein (HDL)), a blood urea test, creatinine, sodium, potassium, magnesium, chloride, ionized calcium, immunoglobulin E (IgE), and thyroid stimulating hormone (TSH).

### 2.5. Statistical Analysis

The participants’ gender is presented as a number and percentage and age is described with median and minimum–maximum values. The distribution of the analyzed blood results was tested using the Kolmogorov–Smirnov test. Since some indicators significantly deviated from the normal distribution, to simplify the presentation, all results of the complete blood count and biochemical analysis for all genotypes are presented as median, minimal, and maximal values. The Mann–Whitney U-test was used to detect significant genetic models. A difference was considered statistically significant if *p* < 0.05. Statistical data analysis was performed using the statistical package IBM SPSS Statistics for Windows, Version 20.0.

## 3. Results

### 3.1. Initial Grouping and Screening

Out of the 370 participants, there were 83 males (22.4%) and 287 females (77.6%). The median age was 40, with a range of 19 to 74 years.

Rs polymorphisms (SNPs) were grouped based on their main qualities and functions related to the skin for further structured evaluation (Table 1).

The SNP frequencies for *AHR* (rs2066853) and *SLC45A2* (rs26722) gene minor variants (MT) were found to be less than 1.00% (Table 2); similar findings were published by De Sousa and colleagues related to cancer patients [19]. A total of 368 subjects were tested for (rs2066853) because two samples were excluded from genotyping due to preanalytical errors. The calculated genotype frequency was found to be similar compared with the allele frequencies from the European population, which is stated as the estimated MAF of each SNP (Table 2).

### 3.2. Genetic Model Analysis and Genotype Association with Routine Blood Screening Parameters

The multiple associations of blood count and immunochemical laboratory parameters with different genotyping models of various polymorphisms were further evaluated for clinical significance related to the actual reference range by considering the calculated median range of each parameter across the population. Only the results with a significance of *p* < 0.05 are presented for demonstration and further analysis. The different genetic models were C1 (WT vs. HT) and C2 (WT vs. MT)—codominant 1 and 2, respectively; D (WT vs. HT + MT)—dominant; and R (MT vs. WT + HT)—recessive. They showed significant values with different alleles for genotypes and models within possible combinations and blood screen results. For example, the C2 (WT vs. MT) model for SNP rs7787362 corresponding to *ELN*—which is related to skin elasticity and support, with *p* = 0.015 and representing a basophile median count of 0.5 for WT and 0.4 for MT genotypes identified within obtained range of actual values—demonstrates a strong correlation between the particular genotype and basophile count (Table A1 and Table A2).

The calculated associations between different parameters of blood count and immunochemistry with corresponding polymorphisms are present in Table 3 and Table 4. Associations with *p* ≥ 0.05 were excluded from the analysis, as they were considered not statistically significant.

Different colors show the variability of associations between particular genotypes with selected parameters of blood screen related to different skin functions. Green cells indicate the normal value (within a reference range) of the blood parameter with the corresponding genotype, which was found to be HT (a pair of heterozygous alleles) or MT (a pair of recessive alleles), demonstrating a positive or neutral effect for the relevant skin property. Purple cells show abnormal, higher values (outside the referenced range) of the blood screen parameter associated with a possible negative effect on the related skin function. For example, the heterozygous genotype (HT) of SNP 1800795, i.e., the IL6 gene, which is related to skin elasticity, is associated with an elevated RBC count. 

Our results of the routine blood screening show no significant associations with the wild-type (WT) genotype of each corresponding SNP, indicating that the population with both wild-type alleles was considered neutral with respect to skin aging.

## 4. Discussion

We carried out a study to examine possible associations between specific genotypes and health parameters, with a focus on skin aging. Recently performed genome-wide association studies (GWASs) and other original research works laid the groundwork for a personalized approach to healthcare, where a niche for more effective skincare approaches certainly exists [20,21,22]. In the literature, there are different types of investigations with different objectives, where some of them are orientated to a thorough pathophysiological and genetical analysis [3,22], while others are focused on a more practical approach, trying to find the best way to utilize scientific findings for clinical use [6,10,11]. Our work stands out from both categories by combining routine laboratory investigations with molecular genotyping results offering regular blood tests that would depend on individual skin-associated genotypes. 

We found significant correlations with liver and pancreatic enzymes, lipoproteins, electrolytes, and blood count parameters with various polymorphisms of genes thought to be involved in skin aging pathogenesis. These results highlight some possible pathways and mechanisms that would explain the resulting phenotypes. Fifteen different genotypes had different associations with blood results, but some tendencies were noticed. Certain genetic variants, namely, CAT (rs1001179), GPX1 (rs1050450), NQO1 (rs1800566), IL1Beta (rs1143634), and COL1A1 (rs1800012), which are primarily associated with antioxidant functions, exhibited significant correlations with liver enzymes ALT and AST, as well as white blood cell counts, particularly with respect to neutrophils (Neu), eosinophils (Eo), lymphocytes (Lymph), basophils (Baso), and monocytes (Mono). These associations highlight their potential roles in maintaining liver health and immune system support. The associations found in our study are consistent with the results of other similar studies, where gene function and SNP associations suggest a multifactorial approach through different mechanisms or platforms and have been discussed in common pathophysiological pathways [11,13,17].

*CAT* (the catalase gene) is known as an important antioxidant enzyme that breaks down hydrogen peroxide to oxygen and water, lowering the impact of ROS (reactive oxygen species) alleviating the negative effect of carcinogenesis, and providing skin protection against radiation and UV [23]. Rs1001179 polymorphism is related to oxidative stress and is observed in chronic hepatitis patients [24]. Our results confirmed that an elevated AST concentration is associated with an rs1001179 minor homozygous genotype, and this finding probably has a negative effect on the skin by lowering the protection against oxidative stress.

The GPX1 (glutathione peroxidase) enzyme is related to antioxidative capacity. The *GPX1* rs1050450 minor homozygous genotype has been implicated in reduced skin antioxidative capacity [11] and was found to be related to various breast, lung, prostate, and colorectal cancers [25]. Its heterozygous and homozygous combination of alleles is associated with elevated Leu, Mo, and Eo levels within inflammation or hyperreactivity pathways. This effect is possibly related to GPX1’s role in removing intracellular hydrogen peroxide, which protects endothelial stability. A *GPX1* mutation with lowered activity was found to accelerate thrombosis and has been associated with stent stenosis [26]. This mechanism probably could work not only in suppressing antioxidative function but also in stiffening endothelial structure in the derma [27]. We speculate that this might have a positive effect on skin sculpture by supporting a stronger base for all epidermic layers. A similar effect is seen after PRP (plasma-rich platelet) injections, releasing a cumulative effect on the skin, besides the stimulation of growth factor release, where indirect NO (nitric oxide) stability and a stiffened endothelium are the result of thrombocyte activation [28,29].

A similar effect is seen with *Il1β* rs1143634, which is known for its activity in skin immunity, and was also found to be involved in cancer development, where it stimulates activated blood monocytes and tissue macrophages [13]. Proinflammatory cytokine Il1β affects cell proliferation, differentiation, and apoptosis, and oxidative stress stimulates an excess release of Il1β, subsequently affecting pancreatic beta cells, which has been implicated in pancreatic cancer development [13]. Based on our findings, we identified a strong association between elevated pancreatic amylase levels and several other factors, which altogether support the theory of inflammation having an overall impact. Interleukins are involved in pyroptosis and inflammasome pathway regulation via transcriptional and translational mechanisms, balancing the normal activity of inflammation, while undue activation influences inflammatory, metabolic, and oncogenic disbalance [30]. The rs1143634 heterozygous genotype was reported to be positively associated with the modulation of inflammasomes, which probably has a protective effect on lung fibroblasts [31] and was found to be associated with dermal fibrosis, which explains why chronic collagen overexpression might be involved in skin aging pathogenesis [32]. 

*NQO1* (the NADPH oxidoreductase gene) is linked to various theories of carcinogenesis, particularly rs1800566, which was found to demonstrate a strong association with gastric cancer or hepatocellular and renal carcinoma through changes in redox status inside the cells [33]. Rs1800566 is known for the same skin antioxidant capacity, along with an expressed reduction in enzymatic activity of the corresponding protein [11]. The predominance of the wild-type genotype, in contrast with the heterozygous alleles of this polymorphism, is linked to increased levels of ALT, CRP, MTL, Baso, and CRE, indicating the influence of common pathways affecting cellular membranes and reduced enzyme activity [33]. Conversely, the minor genotype may exhibit an opposing effect, leading to a reduction in enzymatic function.

The cytokine TNF-α is known to play a role in melanocyte apoptosis, and its polymorphism rs1800629 was found to be associated with immune skin protection. Furthermore, its affected promoter may reduce this ability, therefore correlating with obesity and an increased plasma insulin concentration [16]. We found a strong association of heterozygous genotypes with elevated Lymph, Er, and CRP levels. Recently, its heterozygous genotype was found to be associated with lymphoblastic leukemia [34]. Published studies revealed a direct association between rs1800629 and premature aging due to TNF-α synthesis defect [35], which is probably related to a lack of collagen turnover [17].

Superoxide dismutase 2 (*SOD2*) gene) rs4880 has a significant impact on telomere length, *ELN* rs7787362 is found to be related to prolonged age respondents, and *SLC45A2* haplotypes rs26722 and rs16891982 encode membrane-based proteins that are involved in melanin synthesis [36]. We found these four SNPs in different groups for skin features, but they were all related to elevated TP, Ca, and CRE depending on different genotypes and, of course, the different pathways that should have been involved that affect the overall outcome. It is worth highlighting their most common protective and supporting functions. Rs26722 and rs16891982 are associated with freckle, eye, hair, and skin pigmentation, playing a protective role for skin, and the minor allele of rs16891982 is strongly associated with black hair color [5]. The *ELN* rs7787362 minor allele was found to be associated with striae formation [15] and in similar studies, the *COL1A1* rs1800012 minor allele was speculated to be associated with skin wrinkle formation, as it was found in relation to soft tissue malfunction [37].

The investigation of other genetic variants, specifically the *MMP1* rs1799750 major genotype compared with the heterozygous pair of alleles, revealed an association with increased levels of ALT, Cre, CRP, and IgE. On the other hand, the AHR rs2066853 wild-type genotype was associated with elevated mono, total cholesterol, and MTL levels. These diverse associations within various pathways highlight the extensive involvement of genes and their mutations in multiple functions, particularly related to elasticity, support function, and other physiological processes. Such findings underscore the comprehensive responsibilities of genes in contributing to various biological functions and their potential impact on health outcomes.

Our study involved a stepwise analysis, starting with the genotyping of polymorphisms and associations with subsequent blood count and immunochemistry results. Moreover, this study demonstrated an essential approach to each patient’s individual combination of multiple tests, as it is not enough to only obtain results from a laboratory; further assessment and a comprehensive evaluation by a laboratory medicine doctor, dermatologist, geneticist, or other qualified specialist is required. By compiling information on specific genotypes with results from laboratory testing, a more complete and more accurate picture of overall health is acquired. Further research should be carried out to create an algorithm that would make an evaluation of a large amount of data easier and simpler in routine practice.

It is also important to note that this study had certain limitations. The cross-sectional design limited the ability to demonstrate causal relationships between genotypes and skin aging markers. Additionally, the study population consisted of healthy individuals, which does not represent the entire general population. Further research involving larger and more diverse populations, as well as longitudinal studies, would be valuable in validating and expanding upon these findings, as well as ruling out the effects of any possible unforeseen confounding variables.

## 5. Conclusions

In conclusion, this cross-sectional study shed some light on the possible role of genetic polymorphisms in skin aging by revealing possible associations between certain SNPs and routine blood tests within a skin properties segment. Answering the following questions should help to guide further analysis and understanding of an extended view of the one-segment example: What effect could blood changes have on particular skin aging statuses? Is it possible to modify the weakness of one’s genetic signature by improving the parameters of related pathogenetic mechanisms? These findings demonstrate and support the personalized medicine approach for aging and further add to the growing literature. Further investigation is warranted to fully understand the complex interplay between genetic factors, environmental influences, and skin aging.

## Figures and Tables

**Table 1 jpm-13-01305-t001:** Grouped polymorphisms and their skin-related functions, (see “Single nucleotide polymorphism detection”) and data source.

Antioxidative	Protective	Elasticity and Support	Immune Response	Skin Hydration
*SOD2* (rs4880) [11]	*TYR* (rs1126809) [12]	*MMP1* (rs1799750) [11]	*IL1Beta* (rs1143634) [13]	*AQP3* (rs17553719) [11]
*GPX1* (rs1050450) [11]	*SLC45A2* (rs26722) [14]	*ELN* (rs7787362) [15]	*TNF-α* (rs1800629) [16]	
*NQO1* (rs1800566) [11]	*SLC45A2* (rs16891982) [12]	*COL1A1* (rs1800012) [17]		
*CAT* (rs1001179) [11]	*AHR* (rs2066853) [18]			
*IL6 (rs1800795)* [11]				

**Table 2 jpm-13-01305-t002:** Calculated genotype frequency within the tested Lithuanian population.

Function	Corresponding Gene with Investigated Polymorphism	WT	HT	MT	Estimated MAF	N
Antioxidative	SOD2 (rs4880)	24.86%	49.46%	25.68%	0.49	370
	GPX1 (rs1050450)	53.51%	38.65%	7.84%	0.32	370
	NQO1 (rs1800566)	65.41%	30.27%	4.32%	0.20	370
	CAT (rs1001179)	59.73%	35.95%	4.32%	0.20	370
Protective	TYR (rs1126809)	60.82%	34.86%	4.32%	0.25	370
	SLC45A2 (rs26722)	96.22%	3.78%	0%	0.04	370
	SLC45A2 (rs16891982)	94.05%	5.95%	0%	0.13	370
Elasticity and support	MMP1 (rs1799750)	34.32%	46.49%	19.19%	0.48	370
	ELN (rs7787362)	32.97%	46.76%	20.27%	0.49	370
	COL1A1 (rs1800012)	75.95%	22.70%	1.35%	0.16	370
	AHR (rs2066853)	85.05%	14.68%	0.27%	0.12	368
	IL6 (rs1800795)	28.90%	45.00%	26.10%	0.36	370
Immune response	IL1Beta (rs1143634)	54.32%	36.49%	9.19%	0.22	370
	TNF-α (rs1800629)	78.11%	20.27%	1.62%	0.15	370
Hydration	AQP3 (rs17553719)	49.19%	41.89%	8.92%	0.25	370

WT—wild-type genotype; HT—heterozygous genotype; MT—minor genotype; estimated MAF—MAF for the European population, data from https://www.ncbi.nlm.nih.gov/snp (accessed on 2 June 2017); N—number of samples investigated.

**Table 3 jpm-13-01305-t003:** Associations of complete blood count results with different genotypes.

Function	Antioxidative				Protective			Elasticity					Immune Response		Hydration
CBC Parameter	rs4880	rs1050450	rs1800566	rs1001179	rs1126809	rs26722	rs16891982	rs1799750	rs7787362	rs1800012	rs2066853	rs1800795	rs1143634	rs1800629	rs17553719
RBC												HT		MT	
WBC, total				HT								HT		HT	
LYPH (N)															HT
LYMPH, (%)	MT			MT						MT			MT	HT	HT
EO (N)													HT		
EO (%)		HT											HT		
Baso (N)			MT						MT						
			HT												
Baso (%)			MT						MT						
			HT												
MONO (N)				HT											HT
MONO (%)		HT									HT		MT		
NEUT (N)		HT		MT								HT		HT	
NEUT (%)		HT		MT						HT			MT		HT
PCT					HT										
MPV					HT					HT				HT	

RBC—red blood cell, WBC—white blood cell, N—absolute count, %—relative count, LYMPH—lymphocyte, EO—eosinophil, Baso—basophil, MONO—monocyte, NEUT—neutrophil, PCT—platelet crit, MPV—mean platelet volume, HT—heterozygous genotype, MT—minor genotype.

**Table 4 jpm-13-01305-t004:** Associations of immunochemistry results with different genotypes.

Function	Antioxidative				Protective			Elasticity					Immune Response		Hydration
Immunochemistry Parameter	rs4880	rs1050450	rs1800566	rs1001179	rs1126809	rs26722	rs16891982	rs1799750	rs7787362	rs1800012	rs2066853	rs1800795	rs1143634	rs1800629	rs17553719
Potassium						HT									
Calcium	HT					HT	HT		HT						HT
Calcium, ionized	HT														
Chloride						HT	HT			HT					
Alkaline phosphatase	HT								HT						
Urea	HT									MT		MT			
Pancreatic amylase				HT									MT		
Creatine			HT					HT							
CRP		HT	HT					MT		MT			HT	HT	
								HT		HT					
LDL			HT								HT		HT		
HDL					HT							HT			
Cholesterol											HT	HT			
AST						HT	HT						MT		HT
ALT		HT	MT	MT	HT			HT					MT		
GGT						HT							MT		
													HT		
IgE								MT				MT	MT		

HT—heterozygous genotype, MT—minor genotype, CRP—C-reactive protein, LDL—low-density lipoprotein, HDL—high-density lipoprotein, AST—aspartate aminotransferase, ALT—alanine transaminase, GGT—gamma-glutamyl transferase.

## Data Availability

Study data are available upon request by email: vaiva.patamsyte@lsmuni.lt.

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
