# Peer review of "Genetical Signature—An Example of a Personalized Skin Aging Investigation with Possible Implementation in Clinical Practice"

_jpm, 2023, doi:10.3390/jpm13091305_

Round 1

Reviewer 1 Report

Review comment

In the present manuscript, it indicates the possibility that the composition of blood is associated with SNPs extracted by several GWAS related with skin aging. However, this paper is not well-described. I recommend that this paper be accepted after major revision.

1. In this study, it indicated the composition of blood may be associated with SNPs extracted by several GWAS related with skin aging. However, it did not show the association between the skin condition and SNPs. It had better to show the association the skin condition and SNPs.

2. Does it mean that administering particular blood components to the person with the particular indicated SNPs is good for the skin in the personalized medicine? Does the gene itself may affect the skin directly?

3. In the result part, the results were not well-described.

4. It is difficult for me to understand what the minor genotype (MT) was showing. Is it the ratio of the person with homozygous genotype or somatic mutation?

5. It had better to show the registered minor allele frequency (MAF) of individual 15 SNPs.

Author Response

Dear Reviewer,

many thanks for your revision and valuable comments.

Reviewer 2 Report

Congratulations to the authors for the interesting research. The structure of the manuscript is correct, the content is understandable to the reader. The authors draw the right conclusions from the work on research.

I have a comment:

198 „These findings showed no objections with other studies, where genes functions  and relations within different pathways had been described” The conclusion (choice of words) about laboratory findings is too far-reaching. Many other conditions can affect the values of basic biochemical parameters.

Author Response

Dear Reviewer,

many thanks for your revision and comments.

Round 2

Reviewer 1 Report

Review comment

In the present manuscript, it indicates the possibility that the composition of blood is associated with SNPs extracted by several GWAS related with skin aging. I recommend that this paper be accepted.